# Pretreatment Plasma IL-6 and YKL-40 and Overall Survival after Surgery for Metastatic Bone Disease of the Extremities

**DOI:** 10.3390/cancers13112833

**Published:** 2021-06-07

**Authors:** Michala Skovlund Sørensen, Thomas Colding-Rasmussen, Peter Frederik Horstmann, Klaus Hindsø, Christian Dehlendorff, Julia Sidenius Johansen, Michael Mørk Petersen

**Affiliations:** 1Musculoskeletal Tumor Section, Department of Orthopedic Surgery, Rigshospitalet, University of Copenhagen, 2100 Copenhagen Ø, Denmark; thomas.colding-rasmussen.01@regionh.dk (T.C.-R.); peter.frederik.horstmann@regionh.dk (P.F.H.); Michael.Moerk.Petersen@regionh.dk (M.M.P.); 2Pediatric Section, Department of Orthopedic Surgery, Rigshospitalet, University of Copenhagen, 2100 Copenhagen Ø, Denmark; klaus.hindsoe@regionh.dk; 3Statistics and Data Analysis Danish Cancer Society Research Center, 2100 Copenhagen Ø, Denmark; chrdehl@cancer.dk; 4Department of Medicine, Herlev and Gentofte Hospital, Copenhagen University Hospital, 2730 Herlev, Denmark; Julia.Sidenius.Johansen@regionh.dk; 5Department of Oncology, Herlev and Gentofte Hospital, Copenhagen University Hospital, 2730 Herlev, Denmark; 6Department of Clinical Medicine, Faculty of Health and Medical Sciences, University of Copenhagen, 2730 Herlev, Denmark

**Keywords:** bone metastases, inflammatory response, IL-6, prognostic biomarkers, YKL-40

## Abstract

**Simple Summary:**

Estimating postoperative survival in patients undergoing surgery for metastatic bone disease of the extremities is important in order to choose an implant that will outlive the patient. The present study suggests that plasma IL-6, reflecting the inflammatory state of the patient, is predictive for postoperative overall survival (OS).

**Abstract:**

Background: Plasma IL-6 and YKL-40 are prognostic biomarkers for OS in patients with different types of solid tumors, but they have not been studied in patients before surgery of metastatic bone disease (MBD) of the extremities. The aim was to evaluate the prognostic value of plasma IL-6 and YKL-40 in patients undergoing surgery for MBD of the extremities. Patients and Methods: A prospective study included all patients undergoing surgery for MBD in the extremities at a tertiary referral center during the period 2014–2018. Preoperative blood samples from index surgery were included. IL-6 and YKL-40 concentrations in plasma were determined by commercial ELISA. A total of 232 patients (median age 66 years, IQR 58–74; female 51%) were included. Results: Cox regression analysis was performed to identify independent prognostic factors for OS. IL-6 correlated with YKL-40 (rho = 0.46, *p* < 0.01). In univariate analysis (log_2_ continuous variable) IL-6 (HR = 1.26, 95% CI 1.16–1.37), CRP (HR = 1.20, 95% CI 1.12–1.29) and YKL-40 (HR = 1.25, 95% CI 1.15–1.37) were associated with short OS. In multivariable analysis, adjusted for known risk factors for survival, only log_2_(IL-6) was independently associated with OS (HR = 1.24, 95% CI 1.08–1.43), whereas CRP and YKL-40 were not. Conclusion: High preoperative plasma IL-6 is an independent biomarker of short OS in patients undergoing surgery for MBD.

## 1. Introduction

Surgical treatment strategy for metastatic bone disease (MBD), especially in the appendicular skeleton, depends to some degree upon the residual estimated life expectancy, thus enabling the orthopedic surgeon to provide patient-specific surgical solutions [1]. In most cases the aim is to provide minimal surgical trauma for the patient with an implant that will last for the remainder of the patient’s life without revision surgery or prolonged rehabilitation. Several attempts to predict survival in patients with MBD have been performed [2,3,4,5], but the majority are biased by largely using subjective variables that depended upon the skills of the treating physician. Therefore, the clinical value of these tools is potentially biased. There is therefore a need for objective prognostic variables for prediction of survival in patients undergoing surgery for MBD.

Tumor-promoting inflammation is one of the hallmarks of cancer development and progression [6]. Many proteins (e.g., Interleukin-6 (IL-6) and YKL-40/CHI3L1) secreted by cancer cells, macrophages, neutrophils and fibroblasts stimulate inflammation. IL-6 is a pleiotropic cytokine and plays a major role in angiogenesis, cancer cell survival, chemotherapy resistance and development of liver metastases [7]. Furthermore, IL-6 has a major role in the communication between the cancer cells and the non-malignant cells within the tumor niche [8]. YKL-40 regulates vascular endothelial growth factor and promotes angiogenesis, protects against apoptosis and stimulates tumor progression and metastasis [9]. It has also been suggested that circulating YKL-40 is a biomarker for the functional polarization of tumor-associated macrophages [9]. Several studies [10,11,12,13] have demonstrated that high circulating IL-6 and YKL-40 levels in patients with different types of solid cancer are associated with short overall survival (OS) [14,15]. Plasma IL-6 and C-reactive protein (CRP) have been shown in smaller studies to be prognostic biomarkers for OS in patients undergoing surgery for MBD [16], but the prognostic value of plasma YKL-40 in these patients is unknown. We hypothesized that both plasma IL-6 and YKL-40, at the time of surgery for MBD of the extremities, are prognostic biomarkers for OS, and this was evaluated in the present prospective biomarker study.

## 2. Materials and Methods

A prospective longitudinal study of all patients undergoing surgery for MBD in the extremities at a tertiary orthopedic oncology referral center was conducted from May 2014 to November 2018. The referral center is one of only two highly specialized musculoskeletal tumor centers serving a reliable cross section of the entire Danish population [17]. Due to the social health politics in Denmark all patients referred to and eligible to be treated at a highly specialized center will receive treatment regardless of socio-economic status. Therefore, the population screened for participation into the current study has minimal selection bias.

The patients were invited to participate in the study regardless of surgical solution (implant/no implant, bone resection/no bone resection). In case of multiple surgeries during the inclusion period, only the blood sample from the index surgery was included in the analysis. Patients were included by interview preoperatively, and written consent was obtained. The study was approved by the regional ethic committee (ID no. H-4-2014-005) and the Danish Data Protection agency (ID no. 30-1222).

The inclusion criterion was surgery due to a metastatic bone lesion, including malignant hematological disease of the bone, in the appendicular skeleton including the pelvis. We as orthopedic oncologists do not differentiate between primary cause of malignant lesion when allocating patients into surgical implant, only in terms of estimated residual survival. Thus, patients with hematological disease were included in the current study as the surgical treatment does not differ compared to patients with metastases from solid cancers.

The exclusion criterion was revision surgery for a failed implant. In case of multiple primary surgeries in the study period, the patient was excluded from future study participation after the initial surgery.

### 2.1. Clinical Variables

Performance status of the patients was reported as Karnofsky score [18] and was clinically addressed by a preoperative evaluation by one of the participating doctors (M.S.S., P.F.H. or T.C.). Dissemination status of the cancer disease was evaluated by patient interview combined with evaluation of preoperative imaging (computer tomography (CT), positron emissions tomography (PET), magnetic scans or bone scintigraphy).

Comorbidity: American Society of Anesthesiologists classification (ASA score) [19] was evaluated by the attending anesthesiologist preoperatively. A patient was considered as having ischemic heart disease (IHD) in case of verified echocardiography with cardiac output <45%. Comorbidity with diabetes mellitus (DM) was considered in cases where the patient suffered from this regardless of medical treatment or HbA1C level.

Complete fracture of the treated lesion was evaluated preoperatively by CT or X-ray.

Biochemical variables were obtained prior to surgery. Hemoglobin, alkaline phosphatase, CRP and absolute leukocyte and neutrophile counts were obtained from routine laboratory roundups and were considered missing if they were obtained more than 7 days prior to surgery. Blood samples (plasma) for IL-6 and YKL-40 analysis were obtained separately from the routine blood samples within 3 days prior to surgery. Blood was drawn into a 6 mL ethylene-diamine-tetra-acetic acid tube and stored between 1/2–2 h at room temperature and then centrifuged for 10 min at 3000 rounds per minute. Thereafter plasma was isolated and stored at −80 °C.

Plasma IL-6 was determined by ELISA (Quantikine HS600B, R&D Systems, Abingdon, UK), and plasma YKL-40 was determined by ELISA (Quidel, California, CA, USA) according to the instructions by the manufacturer. The analyses were performed blinded to the clinical data. Elevated plasma IL-6 level was defined as the cut-off for the 95th percentile in healthy blood donors, i.e., >4.50 pg/mL [20]. Since plasma YKL-40 increases exponentially with increasing age, a formula (Percentile = 100/1 + (YKL-40^−3^) * (1.062^age^) * 5000) was used to calculate the age-corrected YKL-40 percentile of the patients [21].

Primary cancer was evaluated by history (as concluded by prior biopsy of primary lesion or imaging) or preoperative biopsy of the bone metastases. To ensure correct diagnosis, histopathology of the lesion was obtained intra-operatively. In case of unknown origin preoperatively (despite preoperative imaging) the primary cause of lesion as shown by intra operative histopathology arbitrated the primary cause for the metastatic lesion. If no primary cancer could be found in histopathology analysis, the primary cancer was considered unknown.

### 2.2. Statistical Analysis

As the present study was explorative, we did not perform power analysis prior to study initiation. To minimize sparse data bias, we chose to dichotomize the following data: (1) ASA score into groups with score 1 + 2 and 3 + 4; (2) Karnofsky score into groups with ≥70 (determined by the patient being self-supportive or not) and <70 and (3) anatomical location in upper and lower extremity.

Since no previous studies have defined a reference interval for plasma IL-6 and YKL-40 in patients with MBD, we decided to dichotomize using the median for IL-6 (11.8 ng/L) and the age-corrected 50% percentile (180 ug/L) for YKL-40 in univariate analysis.

As we expected >20 primary causes for the metastatic lesion to be included into the study, we chose to categorize the variable by prognostic group as proposed by Sørensen et al. [2].

Continuous variables are reported as median and IQR as we did not expect normal distribution, and in order to eliminate the expected right screwed values for biochemical variables, these were reported as log_2_-transformed variables in multivariate analysis.

OS was estimated by the Kaplan–Meier estimator. Correlation between IL-6 and YKL-40 was investigated using the Spearman Rank test. Cox regression models were fitted to identify prognostic factors for OS, and the assumption of proportionality was tested with scaled Schoenfeld residuals.

## 3. Results

During the study period 321 surgeries for MBD of the extremities were performed at our institution. Eighteen patients underwent the second primary procedure during the period, and in 38 cases the surgery was a revision of a failed implant. Thirty-three patients of the remaining 265 patients were lost to inclusion (were not asked to participate). Thus, 232 patients were included in the analysis (Figure 1).

### 3.1. Clinical Characteristics

The baseline clinical characteristics of the patients are shown in Table 1.

The median age of the patients was 66 years (IQR: 58–74 age), and there was an equal distribution between males and females (49%/51%). Eighty-two percent of the patients had multiple metastatic lesions at time of surgery, and 70% of the treated lesions were completely fractured. The patients had poor performance status with 41% of the patients not being able to care for themselves (Karnofsky score ≤70). No lesions showed discrepancy between expected origin of cancer and confirmed primary cause of metastatic lesion in histopathology examination of excised tissue.

The median plasma IL-6 was 12 ng/L (IQR: 5–28), and the median plasma YKL-40 was 131 µg/L (IQR: 68–262). Median levels for other included inflammatory variables were: absolute leukocyte count 7.90 × 10^9^ (IQR: 5.70–10.20); neutrophile count 5.80 × 10^9^ (IQR: 4.00–7.50); CRP 29 mg/L (IQR: 7–67) and alkaline phosphatase 116 U/L (IQR: 79–169).

Patients with solitary lesions and low ASA score (1 + 2) had lower median plasma IL-6 and YKL-40 compared to patients with multiple lesions and high ASA score (Table 2). Plasma IL-6 was higher in patients with moderate or fast growing cancer compared to patients with slow growing cancer (*p* < 0.001), but this was not found for plasma YKL-40 (Table 2).

### 3.2. Plasma IL-6 and YKL-40 and Overall Survival

#### 3.2.1. Univariate Analysis

In univariate Cox regression analysis, increasing levels of IL-6, YKL-40 and CRP (log-transformed continuous variable) were associated with short OS (IL-6: HR = 1.26, 95% CI 1.16–1.37, *p* < 0.01; YKL-40: HR = 1.25; 95% CI 1.15–1.37, *p* < 0.01; CRP: HR = 1.20, 95% CI 1.12–1.29, *p* < 001) (Table 3 and Figure 2).

Figure 3 illustrates Kaplan–Meier plots for patients with low or high YKL-40 (dichotomized according to the age-corrected 50% percentile corresponding to level of 180 ug/L). The median survival was 4.80 months (CI: 3.57–6.77) for patients with high plasma IL-6 and 14.67 months (CI: 10.60–22.50) for patients with low plasma IL-6. The median survival was 6.83 months (CI: 5.07–9.76) for patients with high plasma YKL-40 and 15.77 months (CI: 9.73–28.3) for patients with low YKL-40.

#### 3.2.2. Multivariate Analysis

In multivariate analysis for OS, only high plasma IL-6 (included as log-transformed continuous variables) was prognostic for short OS (IL-6: HR = 1.24, 95% CI 1.08–1.43, *p* < 0.001), whereas other inflammatory variables were not (YKL-40: HR = 0.94, 95% CI 0.82–1.08, *p* = 0.39; CRP: HR = 0.99, 95% CI 0.88–1.12, *p* = 0.86) (Table 4).

Fast growing cancer (HR = 2.80, 95% CI 1.80–4.37, *p* < 0.001), complete fracture of the lesion (HR = 1.69, 95% CI 1.12–2.55, *p* = 0.01), Karnofsky score below 70 (HR = 2.22, 95% CI 1.51–3.26, *p* < 0.001), multiple metastatic lesions (HR = 2.86, 95% CI 1.59–5.16, *p* < 0.01) and visceral metastasis (HR = 1.56, 95% CI 1.07–2.27, *p* = 0.02) were also independent prognostic variables for short OS (Table 4). Hemoglobin (*p* = 0.46), absolute leucocyte count (*p* = 1.00) and neutrophile count (*p* = 0.57) (included in the multivariate analysis as log-transformed continuous variables) and ASA score (*p* = 0.60) were not independent prognostic factors for OS.

Schoenfeld residuals showed increased hazard for IL-6 until ~400–500 days (Figure 4) and a tendency towards a lower hazard ratio initially for YKL-40 that was more neutral (log (HR)~1) as time progressed.

## 4. Discussion

Elevated plasma IL-6 and YKL-40 at the time of diagnosis have been shown to be prognostic biomarkers of short OS in various types of cancer. The present prospective study of 232 patients with MBD in the extremities from various primary cancer types showed that elevated plasma IL-6, but not CRP and YKL-40, was an independent prognostic parameter for short OS.

### 4.1. Limitations

The inclusion of patients in the study was biased by selection of patients treated at a tertiary treatment center, and patients with the poorest expected survival were probably not included in the study. The majority of patients were treated with endoprosthetic reconstructions. There is no evidence to indicate that choice of surgical treatment method influences the survival [17], but the authors cannot exclude such an effect on survival of the population. Previously, we have reported that the magnitude of the surgical trauma was not a risk factor for survival, but the study was weakened by the lack of a comparison group [22].

The categorization of cancer groups into three risk groups may bias the outcome in the current study by introducing misclassification of the actual risk of primary cancer diagnosis and association with IL-6 and YKL-40. We chose to categorize the primary cancer according to a modified Katagiri classification as opposed to using all subgroups in an attempt to minimize sparse data bias, and the Katagiri classification was used because this classification has systematically been used in predictions models for survival over time [2,3,4].

Lastly, the Cox regression model for identifying independent prognostic factors for survival may not have been the optimal choice for analysis as variation of hazard for YKL-40 was found in Schoenfeld plots.

### 4.2. Inflammatory Response

In univariate analysis, we found that plasma IL-6, CRP and YKL-40 were prognostic biomarkers for OS, but in multivariate analysis, adjusting for known risk factors, only IL-6 was an independent parameter. The correlations between IL-6 and CRP and YKL-40 are expected since IL-6 regulates the synthesis of both CRP [23] and YKL-40. However, YKL-40 is not only regulated by IL-6. Other cytokines as well as growth factors and miRNAs have been shown to up- or downregulate YKL-40. Furthermore, stress factors including hypoxia, ionizing radiation and serum depletion can regulate YKL-40 expression and secretion [9].

The inflammatory response may therefore play an important role in disease progression and survival in patients undergoing surgery for MBD in the extremities.

The inflammatory response can be measured in various ways, and in the current study we included neutrophile and leucocyte count, CRP and IL-6. It is well known that a bone fracture induces an inflammatory response [24], and we adjusted for this in the multivariate analysis since 70% of the patients in the present study were treated for a complete fracture. However, IL-6 and bone fractures remained independent prognostic variables. We suggest including both variables in prognostic models in the future. The reason for fracture remaining prognostic may be biased by lead time, as patients not suffering from complete fracture may simply be identified at an earlier cause of their lives than those progressed to complete fracture.

IL-6 is also a myokine, i.e., a protein produced and secreted by skeletal muscles mainly to fulfill paracrine and endocrine roles in the insulin-sensitizing effects following exercise [25]. It is not known how IL-6 produced by skeletal muscles will affect MBD in the extremities.

### 4.3. Performance Status and Comorbidity

It is debatable whether comorbidity impacts survival for this population [26], but performance status seems to be associated with survival. In the current study, we did not address comorbidity directly but rather performance status using ASA score and Karnofsky score as the literature has previously shown a prognostic value for these variables in patients undergoing surgery for long bone metastasis [2,3,5]. The current study showed that the performance status was independently associated with OS as opposed to ASA score. A study of risk factors for 30-day mortality after surgery for MBD in the extremities found both variables to be prognostic [22]. The discrepancy between previous findings and the current study may be explained by ASA score being developed to address pre/perioperative complication rate and mortality and thus not suitable as a prognostic factor for longer survival.

## 5. Conclusions

In conclusion, our study showed that plasma IL-6, bone fracture and Karnofsky score were independent prognostic parameters for OS after surgery for MBD in the extremities. The introduction of IL-6 into prognostic models for OS in these patients may strengthen future models by minimizing subjective bias in current models, but our study cannot support the use of YKL-40 in prognostic models.

## Figures and Tables

**Figure 1 cancers-13-02833-f001:**
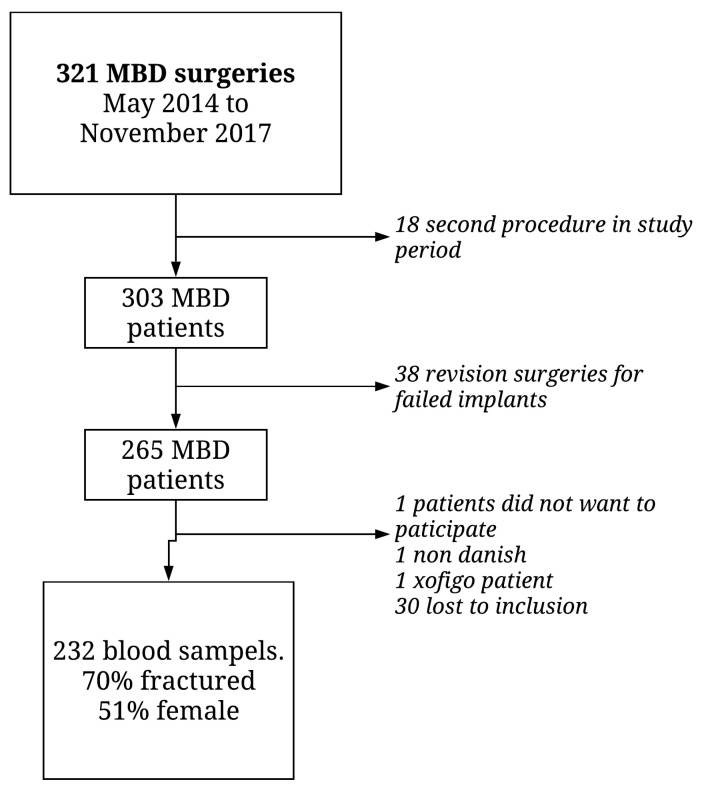
Flow of patients in the study.

**Figure 2 cancers-13-02833-f002:**
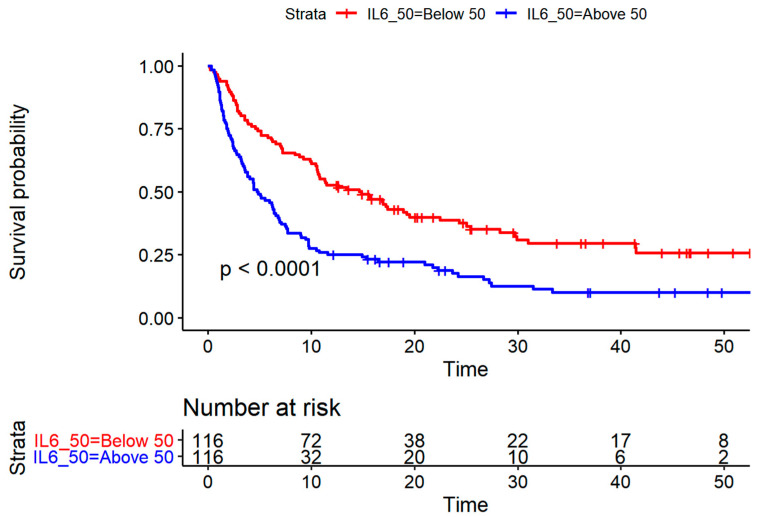
Kaplan–Meier plots for patients with low or high IL-6 dichotomized according to the median level of 11.8 ng/L illustrating better survival probability for patients with low IL-6 (*p* < 0.0001).

**Figure 3 cancers-13-02833-f003:**
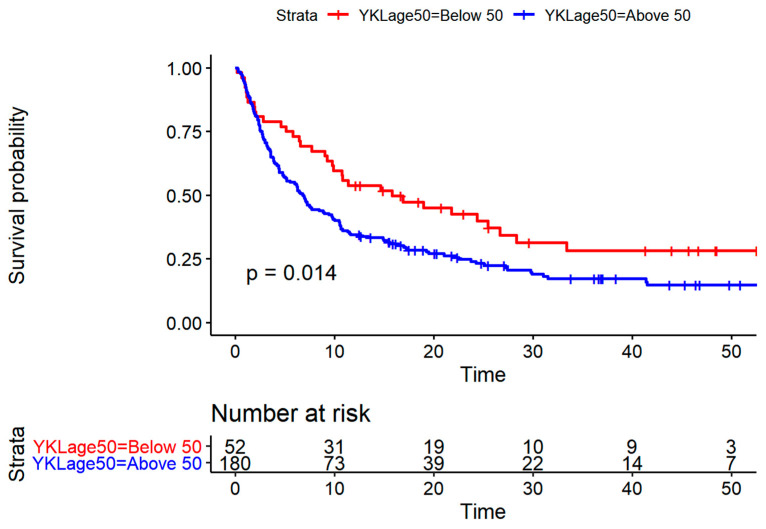
Kaplan–Meier plots for patients with low or high age adjusted YKL-40 dichotomized according to the median level of 180 ug/L illustrating better survival probability for patients with low YKL-40 (*p* = 0.014).

**Figure 4 cancers-13-02833-f004:**
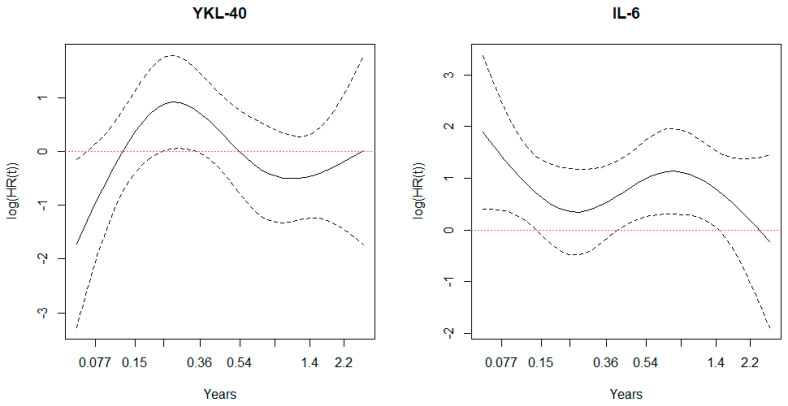
Schoenfeld residuals for hazard over time stratified for IL-6 and YKL-40. The graphs suggest increased hazard over time for IL-6 and decreasing hazard over time for YKL-40, but for both variable change in hazard is observed after 1.5 years after surgery.

**Table 1 cancers-13-02833-t001:** Descriptive data of the cohort (*n* = 232).

Variable	Median (IQR) or *n* (%)
Age, years	66 (58, 74)
Female gender	114 (49.1%)
Cancer:	
Slow growing	73 (31.6%)
Moderate growing	61 (26.4%)
Fast growing	97 (42.0%)
Cancer type (5 most comment)	
Breast	50 (22%)
Lung	46 (20%)
Renal	34 (15%)
Prostate	25 (11%)
Myeloma	23 (10%)
Hemoglobin, mmol/L	7.20 mmol/L (6.50, 8.03)
Leucocyte count × 10^9^	7.90 (5.70, 10.20)
Neutrophil count × 10^9^	5.80 (4.00, 7.50)
C-reactive protein, mg/L	29 (7, 67)
Alkaline phosphatase, U/L	116 (79, 169)
YKL-40, µg/L	131.00 (68, 262)
IL-6, ng/L	11.75 ng/L (4.57, 27.60)
Fracture	163 (70.3%)
Karnofsky < 70	94 (40.9%)
ASA group 3 + 4	131 (56.7%)
Ischemic heart disease (EF < 45%)	16 (6.9%)
Diabetes	29 (12.5%)
Multiple metastatic lesions (visceral + bone)	190 (81.9%)
Visceral metastases present	107 (46.1%)
Lower extremity	190 (81.9%)

**Table 2 cancers-13-02833-t002:** Distribution of plasma YKL-40 and IL-6 concentrations in the different risk groups.

	Plasma YKL-40		Plasma IL-6	
	Median (µg/L) (IQR)	*p*-Value *	Median (ng/L)(IQR)	*p*-Value *
Solitary lesion (*n* = 42)	81 (50, 156)	0.002	7.65 (2.23, 20.30)	0.011
Multiple lesions (*n* = 190)	143 (74, 290)		12.85 (5.30, 29.28)	
ASA 1 + 2 (*n* = 100)	89 (51, 197)	<0.001	9.05 (3.20, 22.60)	0.005
ASA 3 + 4 (*n* = 131)	155 (84, 296)		16.60 (6.20, 29.10)	
Slow growing cancer (*n* = 73)	116 (60, 206)	0.078	7.10 (2.90, 15.40)	<0.001
Moderate growing cancer (*n* = 61)	138 (69, 316)		20.80 (6.20, 31.80)	
Fast growing cancer (*n* = 97)	144 (77, 291)		12.90 (6.10, 27.50)	

* Mann-Whitney test.

**Table 3 cancers-13-02833-t003:** Unadjusted Cox regression model for potential independent prognostic variables for overall survival. All continuous (cont) variables are presented as log_2_-transformed variables.

Variable	Type	HR	2.5%	97.5%	*p*	Reference
Age	Cat.	1.38	1.02	1.85	0.03	<66 year
Gender	Cat	1.22	0.91	1.64	0.19	Female
Moderate growing cancer	Cat	1.28	0.84	1.94	<0.01	Slow growing cancer
Fast growing cancer	Cat	3.14	2.17	4.53	<0.001	Slow growing cancer
Hemoglobin	Cont	0.34	0.19	0.62	<0.01	
Leucocytes	Cont	1.43	1.11	1.83	<0.01	
Neutrophils	Cont	1.34	1.08	1.65	0.01	
CRP	Cont	1.20	1.12	1.29	<0.01	
Alkaline Phosphatase	Cont	1.28	1.11	1.48	<0.01	
YKL-40	Cont	1.25	1.15	1.37	<0.01	
Age-adjusted YKL-40 (>median)	Cat	1.59	1.10	2.32	0.01	89 µg/L
IL-6	Cont	1.26	1.16	1.37	<0.01	
IL-6 (> median)	Cat	1.99	1.47	2.68	<0.01	11.8 ng/L
Fracture	Cat	1.92	1.37	2.71	<0.01	Impending fracture
Karnofsky score	Cat	2.61	1.92	3.56	<0.01	Below 70
ASA	Cat	2.30	1.68	3.16	<0.01	1 + 2
Ischemic heart disease	Cat	1.29	0.75	2.24	0.37	EF < 45%
Diabetes	Cat	0.97	0.62	1.52	0.89	No
Multiple metastatic sites	Cat	3.21	1.98	5.21	<0.01	None
Visceral metastasis	Cat	2.48	1.83	3.35	<0.01	None
Lower extremity	Cat	1.48	0.99	2.22	0.05	Upper extremity

Abbreviations: Cat, Categorical; Cont, continuous; CRP, C-reactive protein; EF, ejection fraction; HR, Hazard ratio; IL-6, interleukin-6.

**Table 4 cancers-13-02833-t004:** Multivariable Cox regression model for potential prognostic variables for overall survival. All continuous variables are presented as log2-transformed variables, i.e., HR corresponds to a fold-change increase, e.g., increase from 2 to 4.

Variable	Category	HR	2.5%	97.5%	*p*	Reference
Age above 66 years	Categorical	1.22	0.86	1.72	0.27	<66 year
Moderate growing cancer	Categorical	1.21	0.75	1.96	0.44	Female
Fast growing cancer	Categorical	2.80	1.80	4.37	<0.01	Slow growing cancer
Hemoglobin (mmol/l)	Continuous	0.74	0.32	1.67	0.46	Slow growing cancer
Leucocyte count	Continuous	1.00	0.34	2.98	1.00	
Neutrophile count	Continuous	1.31	0.51	3.34	0.57	
C-reactive protein (mg/L)	Continuous	0.99	0.88	1.12	0.86	
Alkaline Phosphatase (U/L)	Continuous	0.94	0.78	1.14	0.55	
YKL-40 (µg/L)	Continuous	0.94	0.82	1.08	0.39	
IL-6 (ng/L)	Continuous	1.24	1.08	1.43	<0.01	
Fracture	Categorical	1.69	1.12	2.55	0.01	Impending
Karnofsky score	Categorical	2.22	1.51	3.26	0.00	Below 70
ASA classification	Categorical	1.11	0.76	1.62	0.60	1 + 2
Multiple metastatic lesions	Categorical	2.86	1.59	5.16	<0.01	No
Visceral metastasis present	Categorical	1.56	1.07	2.27	0.02	None

## Data Availability

The data presented in this study are available on request from the corresponding author.

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
