# Peer review of "Pretreatment Plasma IL-6 and YKL-40 and Overall Survival after Surgery for Metastatic Bone Disease of the Extremities"

_cancers, 2021, doi:10.3390/cancers13112833_

Round 1

Reviewer 1 Report

The revised manuscript may be accepted in the present form.

Reviewer 2 Report

Sir, 

I have recently reviewed the re-submitted manuscript "Pretreatment plasma IL-6 and YKL-40 and overall survival after surgery for metastatic bone disease of the extremities" by Michala Skovlund Sørensen and co-workers. 

The authors provided a rebuttal letter reflecting my earlier review. Some issues were modified as suggested - and I am grateful for that ( specifically, the inclusion of primary cancer types).  In some issues, the authors provided sufficient explanation for leaving the manuscript as it was. Alternatively, they provided sufficient background in discussion enabling thus proper understanding and interpretation of their data.  Which I also appreciate. 

Given the relative rarity of the studied disease and the potential significance of IL-6 signalling in cancer, I believe that their manuscript already reached sufficient quality for publication.  I believe that this manuscript can enrich the interested readers. 

I just must highlight that in the presented PDF the Table 1 is full of erroneous signs - I believe this is just an issue of formatting and I do not have to see it again. 

This manuscript is a resubmission of an earlier submission. The following is a list of the peer review reports and author responses from that submission.

Round 1

Reviewer 1 Report

Manuscript Cancers-1195872

In the article “Pretreatment plasma IL-6 and YKL-40 and overall survival after surgery for metastatic bone disease of the extremities”, Michala Skovlund Sørensen and co-workers analyzed plasma levels of IL-6 and YKL-40 in patients undergoing surgery for metastatic bone disease (MBD) to identify independent prognostic factors of Overall Survival (OS). By Elisa assays, the Authors quantified pre-surgical plasma levels of IL-6 and YKL-40 in prospectively enrolled 232 patients to be subject to surgery for MBD. The occurrence of any correlation between IL-6 and YKL-40 levels and performance status (Karnofsky score), anesthesiologist’s classification (ASA score), OS as well as several hematologic parameters, allowed Authors to conclude that unlike C-Reactive Protein and YKL-40, high levels of IL-6 associate with a shorter OS. Since plasma levels of IL-6 reflect the inflammatory state of the patient, the Authors conclude that measurement of plasma levels of IL-6 may be predictive for postoperative overall survival.

The topic is interesting as the identification of prognostic variables predictive of postoperative overall survival may aid surgical oncologist for choosing personalized strategies for patients undergoing surgery for MBD. However, there are a number of concerns that should be addressed.

Major concerns:

The Authors enrolled patients undergoing surgery for MBD from both solid tumors and malignant hematological disease of the bone. Since the behavior of hematological and solid tumors is very different, Authors should discuss criteria adopted to evaluate the IL-6 production in light of impact of an altered bone marrow environment associated to hematological diseases.

Regarding the solid tumors, the Authors never mentioned histological type and grading of primary tumors, neither whether the enrolled patients had undergone chemotherapy or radiotherapy before developing bone metastases. This information should be included in the manuscript and results discussed.

Minor concerns:

The legend of Fig 4 is missing from the text.

The Authors have long last experience in studying the role of YKL-40 in tumor progression and claim that high circulating YKL-40 levels in patients with different types of solid cancer are associated with shorter OS. In light of this statement, a paragraph discussing the possible role of YKL-40 in MBD should be included in the Discussion.

Reviewer 2 Report

Sir, 

I have recently reviewed the manuscript "Pretreatment plasma IL-6 and YKL-40 and overall survival after surgery for metastatic bone disease of the extremities" submitted by Michala Skovlund Sørensen and co-workers to Cancers. The authors present their prospective study on patients undergoing surgery for metastatic bone disease in the extremities at a tertiary referral centre during the period 2014-2018. I believe that the essential message of this manuscript is that high preoperative plasma IL-6 is an independent biomarker of short survival in patients. 

The introduction is brief - however, it presents a sufficient working hypothesis with a minimalistic background. There are numerous reviews available upon this topic (namely IL-6 and YKL-40) available, which can substitute it if properly referenced  (e.g. Brabek et al. 2020  doi: 10.3390/ijms21217937;  and Larionova et al. 2020 DOI:10.3389/fonc.2020.566511 ). 

The study design is proper, and it is practically oriented on the identification of possible biomarkers. However, I must admit that the selection of patients interprets data problematics. Authors describe the selected patients as with "metastatic bone disease", which is suitable for, e.g. imaging studies, but it is very insufficient for cancer biology, biochemistry and biomarker discovery purposes. It is a too gross category. One should not neglect that there are very significant differences between different types of cancer. I believe that authors must reveal the diagnoses (in %) of the presented cohort. It would also be interesting to see the range of IL-6 and YKL-40 in these groups according to primary tumour type. 

Also, the authors included patients with bone involvement and also visceral metastatic lesions. However, it is challenging to draw any conclusion from this. Regrettably, we do not know much about the extraosseous disease burden. The authors included Karnofsky scoring, which is a critical aspect. However, I also miss at least a brief note that IL-6 is btw. important myokine and also depends on physical activity see Pal et al. 2014 DOI: 10.1038/icb.2014.16). This makes the interpretation of observed IL-6 levels somewhat difficult.

The authors also highlighted YKL-40, which is a well known potential biomarker for a while. I believe that it is worthy of attention that chitinase YKL-40 is regulated via STAT3 (please see the evidence summarized at STRING database for deeper insight https://version11.string-db.org/cgi/network.pl?taskId=Pha1LdHtes6Z). It is obvious that these two markers are not independent, and it highlights the IL-6 signalling cascade deregulation. 

To the minor points: Figure 2 - the survival graph requires better graphical artwork. It is also not easily legible; also, the colour code is not optimal. 

To conclude, I highly appreciate the work which was done by the authors. It is worthy of respect and attention. Regrettably, I do not feel that this manuscript is ready. I believe that authors should be encouraged to submit a revised manuscript soon. I am keen to review it again with the above-suggested modifications.